# Application of Nitrate, Ammonium, or Urea Changes the Concentrations of Ureides, Urea, Amino Acids and Other Metabolites in Xylem Sap and in the Organs of Soybean Plants (*Glycine max* (L.) Merr.)

**DOI:** 10.3390/ijms22094573

**Published:** 2021-04-27

**Authors:** Yuki Ono, Masashige Fukasawa, Kuni Sueyoshi, Norikuni Ohtake, Takashi Sato, Sayuri Tanabata, Ryo Toyota, Kyoko Higuchi, Akihiro Saito, Takuji Ohyama

**Affiliations:** 1Graduate School of Science and Technology, Niigata University, Niigata 950-2181, Japan; f15d021h@outlook.jp (Y.O.); sigesige671@gmail.com (M.F.); sueyoshi@agr.niigata-u.ac.jp (K.S.); ohtake@agr.niigata-u.ac.jp (N.O.); 2Faculty of Bioresource Sciences, Akita Prefectural University, Akita 010-0195, Japan; t_sato@akita-pu.ac.jp; 3College of Agriculture, Ibaraki University, Mito 310-0393, Japan; sayuri.tanabata.i@vc.ibaraki.ac.jp; 4Faculty of Applied Biosciences, Tokyo University of Agriculture, Tokyo 156-8502, Japan; 44419013@mpdai.ac.jp (R.T.); khiguchi@nodai.ac.jp (K.H.); asaito@nodai.ac.jp (A.S.)

**Keywords:** amino acid, soybean, ammonia, arginine, nitrate, nodule, urea, ureide

## Abstract

Soybean (*Glycine max* (L.) Merr.) plants form root nodules and fix atmospheric dinitrogen, while also utilizing the combined nitrogen absorbed from roots. In this study, nodulated soybean plants were supplied with 5 mM N nitrate, ammonium, or urea for 3 days, and the changes in metabolite concentrations in the xylem sap and each organ were analyzed. The ureide concentration in the xylem sap was the highest in the control plants that were supplied with an N-free nutrient solution, but nitrate and asparagine were the principal compounds in the xylem sap with nitrate treatment. The metabolite concentrations in both the xylem sap and each organ were similar between the ammonium and urea treatments. Considerable amounts of urea were present in the xylem sap and all the organs among all the treatments. Positive correlations were observed between the ureides and urea concentrations in the xylem sap as well as in the roots and leaves, although no correlations were observed between the urea and arginine concentrations, suggesting that urea may have originated from ureide degradation in soybean plants, possibly in the roots. This is the first finding of the possibility of ureide degradation to urea in the underground organs of soybean plants.

## 1. Introduction

Soybean plants form root nodules with symbiotic soil bacteria such as bradyrhizobium, and they can fix atmospheric nitrogen (N_2_). Soybean plants also absorb N from soil and fertilizers through their roots. However, nodule formation, nodule growth, and N_2_ fixation activities are repressed when the nodulated roots are directly exposed to high concentrations of the combined forms of nitrogen, especially nitrate, a major form of inorganic nitrogen in upland fields [1,2,3,4]. Multiple effects of nitrate inhibition have been reported, such as the decrease in nodule number, mass, and N_2_ fixation activity as well as the acceleration of senescence or disintegration of established nodules, so nitrate inhibition cannot be explained by a simple mechanism [2,5].

As for the initial assimilation of fixed nitrogen in the soybean nodules, previous experiments revealed that ammonia produced by nitrogen fixation in the bacteroid is rapidly released to the cytosol of infected cells and is initially assimilated into the amide group of glutamine (Gln) by the enzyme glutamine synthetase (GS). Glutamine and 2-oxoglutarate (2-OG) produce two molecules of glutamate (Glu) using the enzyme glutamate synthase (GOGAT) [6,7,8]. Some Gln is used for de novo purine synthesis in infected cells, and urate is transported to the adjacent uninfected cells in the central symbiotic region of the nodule [9,10,11]. Urate is catabolized into allantoin and allantoate in the uninfected cells, and then, they are transported to the shoot through the xylem vessels of the roots and stems (Figure 1).

### 1.1. Ureide Biosynthesis in Soybean Plants

Allantoin and allantoate, collectively called ureides, are universal metabolites in animals, plants, and microorganisms, which are generally produced from surplus purine degradation. In plants, these compounds play an important role in the storage and translocation of N in several species such as maple (*Acer saccharum*) and comfrey (*Symphytum officinale*) [12,13]. It has been established that ureides are the principal compounds that transport N from the soybean nodules to the shoot via the xylem vessels, but a small amount of asparagine (Asn) is also transported. It is noteworthy that economically important legume crops such as soybeans, beans (*Phaseolus vulgaris* L.), and cowpeas (*Vigna unguiculate*) are ureide-transporting plants [14]. Regarding ureide synthesis in soybean plants, Kushizaki et al. [15] discovered that ureide concentration in the stems and petioles of nodulating soybean was much higher than that in the non-nodulating isoline during all growth periods. Subsequently, Ishizuka found that nodulated soybean roots transport a large amount of ureides to the shoots, as compared with non-nodulated roots [16]. Ureides are rapidly synthesized in the nodules from fixed ^15^N_2,_ and these are the main compounds transported to the shoot [7,8,17,18]. The same observation that the majority of the fixed nitrogen in soybean nodules is transported to the shoot in the form of ureides was previously published [19,20].

It is true that non-nodulated soybean roots also synthesize ureides and transport them to the shoots, although in much lower amounts than those from the nodules [16]. The specific site and metabolic pathways involved in ureide synthesis in the roots are still uncertain. When ammonium was supplied to the amide transporting type legume, alfalfa (*Medicago sativa*), ureide concentration was higher in the lateral roots and the nodules than in the other tissues. In the main root, ureide concentration was the highest in the root tip [21].

The reaction converting urate to allantoin in nodules is catalyzed by a nodule-specific uricase (urate oxidase). However, Raychaudhuri and Tipton [22] reported a novel ureide-metabolizing enzyme, hydroxyisourate hydrolase, which catalyzes the hydrolysis of hydroxyisourate (HIU), the product of the urate oxidase reaction, to generate 2-oxo-4-hydroxy-4-carboxy-5-ureidoimidazoline (OHCU), an intermediate in the nonenzymatic formation of allantoin from HIU; the decarboxylation and tautomerization of OHCU yield allantoin (Figure 1).

### 1.2. Degradation of Ureides

The hydrolysis of allantoin to allantoate via allantoinase has been observed in the nodules, roots, stems, leaves, and fruits of soybeans [22,23]. For allantoate degradation, there are two pathways in microorganisms via allantoate amidinohydrolase (EC 3.5.3.4) (Pathway A in Figure 1), and allantoate amidohydrolase (EC 3.5.3.9) (Pathway B in Figure 1) [24].

Studies to evaluate which degradation pathway is dominant in soybean leaves have been extensively carried out since 1980. From the results of leaf disc analysis of soybean cultivar, ‘Maple Arrow’ after providing ^14^C-labeled allantoin combined with a urease inhibitor acetohydroxamate (AHA), Shelp and Ireland [23] reported that allantoate is decomposed through pathway A via ureidoglycolate, which leads to producing two urea molecules from one allantoate. Atkins et al. [25] also identified NH_3_, urea, and CO_2_ as ureide breakdown products in cowpea. On the other hand, Winkler et al. [26,27] reported that the leaf extracts and intact leaves of the soybean cultivar ‘Williams 82′ liberated NH_3_ and CO_2_ directly from allantoate without releasing urea intermediates, suggesting that it utilized an amidohydrolase pathway (Figure 1, Pathway B). Later, Todd and Polacco [28] compared the protein fractions and total leaf homogenates from leaves of both cultivars for the ability to evolve either ^14^CO_2_ or ^14^C-urea from ^14^C-labelled ureides in the presence of various inhibitors. They observed no significant biochemical differences in the ureide degradation pathways between these two cultivars. They found that leaf homogenates and leaf discs of both cultivars evolved both ^14^CO_2_ and ^14^C-urea, although pathway B was preferred. The same group reported that urease is not essential for ureide degradation, comparing urease negative mutants and the control soybean plants [29], and they suggested that arginine may be a major precursor of urea in plants rather than ureides.

Werner et al. [30] investigated the ureide-degrading reactions of purine ring catabolism in *Arabidopsis thaliana*, soybean, and rice (*Oryza sativa*), and four enzymes of Arabidopsis, (1) allantoinase, (2) allantoate amidohydrolase (AAH), (3) ureidoglycine aminohydrolase, and (4) ureidoglycolate amidohydrolase (UAH), in catalyzing the complete hydrolysis of the ureide allantoin in vitro. They found that orthologous genes occur in all plant genomes sequenced to date, indicating that the amidohydrolase route (Pathway B) of ureide degradation is universal in plants. However, the metabolic route in vivo remains controversial. Regarding urea formation, they suggested that the allantoate degradation intermediates, ureidoglycine, and ureidoglycolate, rapidly decayed non-enzymatically and released urea and glyoxylate (Figure 1B). Furthermore, Muñoz et al. purified and characterized the enzyme ureidoglycolate urea-lyase from chickpea (*Cicer arietinum*) [31] and French bean (*Phaseolus vulgaris*) [32]. Both enzymes catalyze the degradation of uredoglycolate to urea and glyoxylate. The same group reported that this enzyme requires phenylhydrazine as a substrate for activity and suggested that the N from ureidoglycolate may be transferred to some unidentified endogenous compounds which have structures similar to phenylhydrazine [33].

In the present study, we investigated the effect of the application of combined nitrogen (N), nitrate, ammonium, or urea, on the metabolite concentrations of xylem sap and the plant parts of nodulated and non-nodulated soybeans, and whether different N forms affect the metabolism and transport of various metabolites either from nitrogen fixation by nodules or the N absorption from the roots.

## 2. Results

Nodulated soybean plants were supplied with 5 mM N as sodium nitrate (5 mM), ammonium chloride (5 mM), or urea (2.5 mM) for 3 d from 20 to 23 days after planting (DAP), and the effects of nitrogen supply on the concentrations of ureides, urea, amino acids, inorganic anions, sugars, and organic acids in the xylem sap and each plant part was analyzed 23 DAP. The control plants were grown with an N-free culture solution throughout the experimental period. The present experiments used hydroponics without soil microorganisms; therefore, urea may be directly absorbed by the roots, although plants were not aseptically cultivated. A group of soybean plants was cultivated without bradyrhizobium inoculation, and the non-nodulated plants were cultivated by supplying them with 1 mM nitrate as a sole N source. We used a lower concentration of nitrate for non-nodulated soybeans due to a long supply period from 5 DAP to 23 DAP.

### 2.1. Effects of Supplying Various Nitrogen Compounds on Plant Growth

The volume of xylem sap collected in 1 h after 3 d of N treatments 23 DAP is shown in Figure 2a. The average volume of xylem sap collected in 1 h was 0.124 mL (control), 0.173 mL (nitrate), 0.176 mL (ammonium), 0.199 mL (urea), and 0.622 mL (non-nodulated). The volume in non-nodulated plants was about three times higher than that in the nodulated plants, which might be due to higher plant growth caused by the long-term supply of 1 mM NaNO_3_ after transplanting to hydroponics at 5 DAP.

The average total plant dry weights (Figure 2b) were 1.33 g (control), 1.34 g (nitrate), 1.21 g (ammonium), 1.34 g (urea), and 2.66 g (non-nodulated). The total dry weight of non-nodulated soybeans was almost twice that of the nodulated plants under the other treatments. Continuous nitrate supply to non-nodulated soybeans from 5 to 20 DAP enhanced plant growth unless nodules were absent. Among the nodulated plants, ammonium-treated plants showed relatively lower dry weight, although the difference was not statistically significant.

The dry weights of the roots, stems, and leaves of non-nodulated plants were significantly (*p <* 0.05) higher than those of nodulated plants with all treatments. The dry weights of roots, stems, and leaves were similar among the nodulated plants undergoing control, nitrate, ammonium, and urea treatments for 3 d. Average dry weight values for nodules were 10.7 mg (control), 9.84 mg (nitrate), 8.83 mg (ammonium), and 9.42 mg (urea). Based on Tukey’s test, the nodule dry weights of the nitrate- and ammonium-treated plants were significantly different (*p <* 0.05) as compared to those of the control and urea-treated plants.

### 2.2. Inorganic Compositions after Application of Various Chemical Forms

The nitrate concentrations in the xylem sap (Figure 3A(a)) and each part (Figure 3A(b)) of the soybean plants after 3 days of treatment are shown. Nitrate was not detected in the xylem sap or plant organs with control, ammonium, and urea treatments, suggesting that nitrification in this hydroponic system was negligible. The concentrations of nitrate in the xylem sap of nitrate-treated plants with 3 d of 5 mM nitrate (16.9 mM) and non-nodulated plants cultivated with 1 mM nitrate (17.6 mM) were similar (Figure 3A(a)), although the concentrations of nitrate in the leaves and roots were relatively higher in the 5 mM nitrate-treated nodulated plants than in the 1 mM nitrate-treated non-nodulated plants (Figure 3A(b)).

The ammonium concentrations in the xylem sap (Figure 3B(a)) were about 0.5–0.8 mM and much lower than the nitrate concentrations in the nitrate-treated plants (Figure 3A(a)). Among the various treatments, the concentrations of ammonium in the xylem sap, roots, and stems subjected to the ammonium treatment were higher than those of the other treatments, although the average concentrations in the leaves and nodules were relatively similar among the treatments, including non-nodulated plants (Figure 3B(b)). However, the ammonium concentration, expressed as μmol per g of dry weight (μmol/g DW), in the nodules with nitrate treatment was 4.3 μmol/g DW, which was significantly lower than that of the other treatments such as the control (7.5 μmol/g DW), which was possibly due to nitrate repression on the nitrogen fixation.

The phosphate concentrations in the xylem sap were relatively high, about 2.5–5.5 mM, while the initial phosphate concentration in the culture solution was 0.05 mM (Appendix A). The application of ammonium and urea significantly increased the phosphate concentration in the xylem sap compared to the control. On the other hand, the xylem sap obtained from the nitrate-treated and non-nodulated plants had decreased phosphate concentrations. In the roots (Appendix A), the phosphate concentrations in the control (99 μmol/g DW) and ammonium (112 μmol/g DW) treatments were higher than those of the nitrate-treated (62 μmol/g DW) and non-nodulated (13 μmol/g DW) plants. The average phosphate concentration in the nodules with the nitrate (18 μmol/g DW), ammonium (12 μmol/g DW), and urea (23 μmol/g DW) treatments was higher than that of the control (2.1 μmol/g DW) plants.

The sulfate concentration in the xylem sap was about 0.5–2.2 mM (Appendix A). The application of ammonium significantly increased the sulfate concentration in xylem sap as compared with the nitrate treated and non-nodulated plants. The average sulfate concentration in each part was relatively similar among the treatments (Appendix A).

The chloride concentration in the xylem sap was about 0–1.5 mM (Appendix A). Application of ammonium chloride significantly increased the chloride concentration in the xylem sap as compared with those of the control and urea treatments. Chloride was not detected in the xylem sap of nitrate treated and non-nodulated plants. The average chloride concentration in each part was relatively higher in the ammonium chloride-treated plants than in the other treatments (Appendix A). The concentration of chloride in non-nodulated roots was the highest (255 μmol/g DW) among the treatments.

### 2.3. Ureides and Urea Composition

The average ureide concentration in the xylem sap was about 0.5–10 mM (Figure 4A(a)). Ammonium (5.7 mM) and urea (5.1 mM) application significantly decreased the ureide concentration in the xylem sap compared with that of the control (9.7 mM). Furthermore, xylem sap from nitrate-treated and non-nodulated plants had very low ureide concentrations. The ureide concentration was similar between the nitrate and control treatments in the leaves, stems, and roots, but was different in the nodules for the control (18.3 μmol/g DW) and nitrate treatment (6.0 μmol/g DW) (Figure 4A(b)). The ureide concentrations in the leaves and nodules after ammonium treatment were much higher than those of the control plants. The ureide concentrations in the leaves and roots after urea treatment were higher than those in the control. The non-nodulated plants, which received 1 mM nitrate as the N source, showed higher growth activity than the nodulated plants, so these plants should have contained more N. The non-nodulated plants had the lowest ureide concentration due to the lack of nitrogen fixation, but a small amount of ureides were present in all organs among the treatments.

The urea concentration was about 0.3–2.2 mM in the xylem sap (Figure 4B(a)). Ammonium treatment (1.56 mM) and urea treatment (1.93 mM) did not change the urea concentration in the xylem sap compared with the control (2.19 mM). On the other hand, xylem sap from nitrate treated (0.62 mM) and non-nodulating (0.26 mM) plants showed decreased urea concentrations at very low levels. The urea concentrations (μmol/g DW) were similar between nitrate and control treatments in leaves, stems, and roots but different in the nodules with the control (9.4 μmol/g DW) and nitrate (1.85 μmol/g DW) treatments (Figure 4B(b)). Urea concentrations in the leaves (8.16 μmol/g DW), stems (14.9 μmol/g DW), and roots (6.91 μmol/g DW) for the ammonium treatment were much higher than those of the control plants. Urea concentrations in leaves (4.08 μmol/g DW), stems (9.21 μmol/g DW), and roots (11.7 μmol/g DW) for the urea treatment were higher than those of the control. The urea concentration in the non-nodulated plants was the lowest for all organs among the treatments.

The average Gln concentration in the xylem sap was about 0.2–3 mM (Figure 5A(a)). The application of ammonium (2.9 mM) and urea (1.8 mM) significantly increased the Gln concentrations in the xylem sap compared to the control (0.68 mM). On the other hand, the xylem sap from the nitrate treated (0.13 mM) and non-nodulated plants (0.18 mM) had decreased Gln concentrations at low levels. The glutamine concentrations were similar between the nitrate and control treatments for the leaves, stems, and roots, but varied in the nodules for the control (4.9 μmol/g DW) and nitrate (1.5 μmol/g DW) treatments (Figure 5A(b)). The glutamine concentrations in the leaves, stems, roots, and nodules after the ammonium treatment were much higher than those of the control plants. The glutamine concentrations in the stems and roots (96 μmol/g DW) subjected to the urea treatment were markedly higher than those of the control. The glutamine concentrations in the non-nodulated plants were similar to the control for all the organs.

The Asn concentration was approximately 2–10 mM in the xylem sap (Figure 5B(a)). The application of ammonium (9.3 mM) and urea (7.6 mM) significantly increased the Asn concentration in the xylem sap as compared with that of the control (2.4 mM), similar to Gln (Figure 5A(a)). On the other hand, xylem sap from the nitrate treatment (3.8 mM) and non-nodulated (5.2 mM) plants had increased Asn concentrations relative to those of the control. The asparagine concentrations in all the organs of the nitrogen treatments were higher than those for the control treatment (Figure 5B(b)). The asparagine concentrations in the leaves, stems, and roots increased but they decreased in the nodules subjected to the nitrate treatment as compared with those of the control. The asparagine concentrations in the leaves (11.5 μmol/g DW), stems (73 μmol/g DW), roots (55 μmol/g DW), and nodules (31 μmol/g DW) for the ammonium-treated plants were much higher than those of the control plants. The asparagine concentrations in the stems (53 μmol/g DW) and roots (86 μmol/g DW) for the urea-treated plants were also markedly higher than those of the control. The asparagine concentration in the non-nodulated plants was higher than that of the control plants for xylem sap and all the organs.

The average Glu concentrations in the xylem sap were relatively constant, at about 0.44–0.60 mM (Figure 5C(a)), and there were no significant differences among the treatments. The glutamate concentrations in all the organs of the control treatment were the lowest among all the treatments (Figure 5C(b)). The glutamate concentrations in the leaves, stems, and roots were slightly increased by the nitrate, ammonium, and urea treatments as compared with those of the control, but were similar in the nodules. The glutamate concentrations were higher in the nodules than in the leaves, stems, and roots irrespective of the treatment.

The average aspartate (Asp) concentrations in the xylem sap were about 0.12–0.90 mM (Figure 5D(a)), the lowest in the control (0.12 mM), and high in the ammonium (0.90 mM) and urea (0.90 mM) treated plants, followed by nitrate (0.55 mM). The Asp concentration in the non-nodulated plants was low (0.16 mM) and not significantly different from the control. The aspartate concentrations in the leaves, stems, roots, and nodules were similar among all the treatments (Figure 5D(b)). The Asp concentration was higher in the nodules than those in the leaves, stems, and roots, irrespective of the treatment. The level of Asp concentration in each organ was similar to that of Glu (Figure 5C(b)).

The average arginine (Arg) concentrations in the xylem sap were relatively low, about 0.09–0.28 mM, and the concentration was lowest in the non-nodulated plants (Figure 5E(a)). The Arg concentration in the control leaves was 0.1 μmol/g DW, and it drastically increased in the leaves subjected to the ammonium and urea treatments (Figure 5E(b)). The concentration of Arg in the nodules was increased by the urea treatment.

The results for the other amino acids are shown in the Appendix A. The xylem sap concentrations of glycine (Gly) were very low, about 0.01–0.02 mM (Appendix A), and those of serine (Ser) (Appendix A), alanine (Ala) (Appendix A), and γ-aminobutyrate (GABA) (Appendix A) were about 0.1–0.2 mM, with no significant differences among the treatments. The Gly concentration in the nodules was several times higher than that of the leaves, stems, and roots, and Gly concentrations in the nodules decreased when subjected to the nitrate, ammonium, or urea treatments compared to the control. (Appendix A). The concentrations of Ser (Appendix A), Ala (Appendix A), and GABA (Appendix A) in the roots were increased in the nitrate-, ammonium-, and urea-treated plants, and in non-nodulated plants, although the concentrations in the nodules for the nitrate treatment decreased as compared with those of the control plants.

The patterns of the amino acid concentrations in the xylem and plant parts were relatively similar for proline (Pro) (Appendix A), tyrosine (Tyr) (Appendix A), phenylalanine (Phe) (Appendix A), lysine (Lys) (Appendix A), leucine (Leu) (Appendix A), isoleucine (Ile) (Appendix A), valine (Val) (Appendix A), and threonine (Thr) (Appendix A). The xylem sap concentrations of Pro (Appendix A), Tyr (Appendix A), and Phe (Appendix A) were low (about 0.03–0.07 mM), whereas the concentrations of Lys (Appendix A), Leu (Appendix A), and Ile (Appendix A), Val (Appendix A), and Thr (Appendix A) were about 0.1–0.5 mM, and the concentrations of those amino acids in the xylem sap were significantly higher in the ammonium-treated plants than in the control. The concentrations of these amino acids in the stems were specifically higher in the non-nodulated plants, although the concentrations in the nodules were relatively constant among the treatments (Appendix A). The histidine (His) concentrations in the xylem sap for the ammonium-treated plants were higher than those of the control, similar (Appendix A) to the above amino acids; however, the His concentration in each organ was relatively constant among the treatments (Appendix A). The tryptophan concentrations in the xylem sap were about 0.01–0.02 mM, and the values were not significantly different among the treatments (Appendix A). Tryptophan (Trp) concentrations in the roots and nodules were increased by nitrate, ammonium, and urea treatments (Appendix A). The cysteine (Cys) concentrations were not significantly different among the treatments (Appendix A) and ranged from 0.06 to 0.09 mM. The Cys concentrations in the stems and roots increased in the non-nodulated plants (Appendix A).

### 2.4. Sugar Composition

Figure 6A–D shows the concentrations of sugars in the xylem sap and organs of the soybean plants. Among the carbohydrates in the xylem sap of control plants, the average concentrations of fructose (Frc) (0.58 mM) and glucose (Glc) (0.51 mM) were higher than those of sucrose (Suc) (0.32 mM) and myo-inositol (Ino) (0.24 mM). After 3 d of ammonium and urea treatments and for the non-nodulated plants, the concentration of sucrose in the xylem sap was significantly lower than that of the control plants (Figure 6A(a)). The concentrations of Suc in the plant organs remained relatively constant among the treatments, except for the leaves subjected to nitrate and ammonium treatments and the non-nodulated plants. The concentrations of Glc (Figure 6B) and Frc (Figure 6C) in the xylem sap were relatively constant, except for the ammonium treatment, which was lower than the control. The concentrations of Glc and Frc in each organ among each treatment were very similar. Those in the roots for the nitrate, ammonium, and urea treatments were significantly lower than those of the control roots. The concentrations of Ino in the xylem sap and plant parts were not significantly different among the various nitrogen treatments, although the Ino concentration in the non-nodulated plants was higher than that of the control plants (Figure 6D). The concentrations of Ino were the highest in the nodules (40–50 μmol/g DW), as compared with the roots, stems, and leaves irrespective of N treatments. It may be related that myo-Ino is a precursor of pinitol, being actively produced in the soybean nodule.

### 2.5. Organic Acid Composition

Among the organic acids (Figure 7A–F) in the xylem sap of the control plants, the average concentration of malate (2.51 mM) was the highest, followed by citrate (0.92 mM), malonate (0.27 mM), α-ketoglutarate (α-KG) (0.23 mM), succinate (0.18 mM), and fumarate (0.12 mM). After 3 d of nitrate and ammonium treatments and in non-nodulated plants, the concentrations of citrate (Figure 7A), α-KG (Figure 7B), and malate (Figure 7E) in the xylem sap were significantly lower than those of the control plants.

The concentration of succinate (Figure 7C) and fumarate (Figure 7D) in the xylem sap decreased under the nitrate treatments, but not by the ammonium and urea treatments. Malate concentrations in the plant organs were lower in the leaves, stems, roots, and nodules in plants supplied with ammonium (Figure 7E). The citrate concentrations in the leaves and nodules treated with nitrate were significantly higher than those of the control plants, although the concentrations in all the organs for the ammonium treatment were the lowest among all the treatments (Figure 7A). The concentrations of malonate in the xylem sap were relatively lower than those of malate and citrate, but the concentrations in the leaves and stems were very high at about 50–100 μmol/g DW (Figure 7F).

### 2.6. Ratios of the Metabolite Concentration in the Treated Plants to That of the Control Plants

The ratios of the concentration of N compounds (Appendix A), sugars (Appendix A), and organic acids (Appendix A) in the treated plants and non-nodulated plants compared to the control nodulated plants are shown in Appendix A (N compounds).

In the nitrate-treated plants (Appendix A), the ratios of the concentrations of N compounds in the xylem were the highest for Asp (4.7-fold) and higher by 1-fold for Asn, Glu, and ammonium, although lower for Glu, ureides, and urea. The concentration ratios of Asn, Asp, and Glu were higher by 1-fold in the leaves, stems, and roots. By contrast, the ratios of most compounds in the nodules were lower by 1-fold, including Asn and Asp. It is noteworthy that the concentration ratios of ureides and urea were very low in the nodules and xylem sap, possibly due to the repression of nitrogen fixation.

In the ammonium-treated plants (Appendix A), the concentration ratios of most of the amino acids in the xylem were higher by 1-fold, and were especially high in Asp (7.7-fold), Gln (4.5-fold), and Asn (4.0-fold). However, the concentration ratios of ureides and urea were lower by 1-fold. With this treatment, the concentration ratios of Arg (23.6-fold), Asn (21.2-fold), ureides (4.9-fold), and urea (4.2-fold) in the leaves were higher by 1-fold. Concentration ratios of Asn and Gln in the roots and stems were markedly higher than those undergoing the nitrate treatment. Ratios of N compounds in the nodules were mostly higher by 1-fold in the nodules, and Asn was the highest (4.9-fold). The concentration ratios of ureides and urea were not lower by 1-fold in the nodules treated with ammonium.

For the urea treatments (Appendix A), the concentration ratios of most of the N-compounds were similar to those of the ammonium treatment. The ratios in the xylem sap were especially high in Asp (7.8-fold), Gln (2.8-fold), and Asn (3.2-fold), and the concentration ratios of ureides and urea were lower than 1-fold. With this treatment, the concentration ratios of Arg (16.1-fold), Asn (14.9-fold), ureides (2.6-fold), and urea (2.1-fold) in leaves were higher by 1-fold. Concentration ratios of Asn and Gln in the roots and stems were high and were similar to those of the ammonium treatment. Ratios of N compounds in the nodules were mostly higher by ×1 in the nodules, with Asn as the highest (2.2-fold). However, the ratios of concentrations of ureides and urea were lower by 1-fold in the nodules treated with urea, which was different from the ammonium treatment.

In the case of the non-nodulated plants continuously supplied with 1 mM nitrate (Appendix A), Asn (2.2-fold) showed the highest concentration ratio in the xylem sap, and the ratios were very low for the ureides (0.05-fold) and urea (0.12-fold). The Asn concentration ratios in the leaves (10.2-fold), stems (4.2-fold), and roots (5.4-fold) were also high among the N compounds. The concentration ratios of ureides and urea in the leaves, stems, and roots of non-nodulated plants were lower by 1-fold.

Appendix A shows the ratios of the sugar concentrations in the treated plants to that of the control plants. All treatments decreased the concentration ratios of Suc, Glc, Frc, and Ino in the xylem sap and most plant parts. For the nitrate treatment (Appendix A) the ratios of Glc (0.25-fold) and Frc (0.24-fold) were very low in the roots, and the ratios of Suc (0.74-fold), Glc (0.50-fold), and Frc (0.81-fold) were also low in the nodules. Ammonium (Appendix A) and urea (Appendix A) treatments decreased the concentration ratios of Glc and Frc in the roots. In non-nodulated plants, the concentration ratios of inositol were high in the leaves, stems, and roots, although low in the xylem sap (Appendix A).

For the organic acids (Appendix A), the nitrate treatment decreased the concentration ratios of malic acid (0.15-fold), citric acid (0.25-fold), and malonic acids (0.88-fold) in the xylem sap, although the ratios of citric acid in the leaves (3.7-fold) and nodules (2.0-fold) were higher by 1-fold. Ammonium treatment caused a decline in all the organic acids listed, either in the xylem sap or in the plant parts (Appendix A). Urea treatment showed a similar pattern, but some organic acids showed a ratio of over 1-fold (Appendix A). Non-nodulated plants showed similar ratios to the nitrate treatment, e.g., lower ratios for malate (0.15-fold), citrate (0.41-fold), and malonate (0.81-fold) in the xylem sap, and a higher ratio of citrate in the leaves (2.4-fold).

Appendix A shows the concentration ratios of minor amino acids not listed in Appendix A. In the xylem sap, the ratios tended to be higher by 1-fold, and those of phenylalanine, leucine, isoleucine, valine, and threonine were relatively high. In the leaves, stems, and roots, the concentration ratios of most amino acids were higher by 1-fold, and the ratio of alanine in the roots was the highest for that plant organ.

### 2.7. Correlation between Urea and Ureides

The correlations between urea and ureides concentrations in the xylem sap (Figure 8A), nodules (Figure 8B), roots (Figure 8C), stems (Figure 8D), and leaves (Figure 8E) are shown. Correlations between urea and arginine, which is another candidate for the urea precursor, are also shown together. The positive correlations between the urea and ureides concentrations were observed in the xylem sap (R^2^ = 0.804), leaves (R^2^ = 0.929), stems (R^2^ = 0.651), roots (R^2^ = 0.874), and nodules (R^2^ = 0.705). There were no correlations between urea and arginine in the xylem sap (R^2^ = 0.332), stems (R^2^ = 0.052), roots (R^2^ = 0.169), or nodules (R^2^ = 0.119), although a positive correlation was observed in the leaves (R^2^ = 0.774). These results suggest that the urea in the xylem sap and soybean plants may be related to ureide metabolism and it is possibly a degradation product of allantoate.

### 2.8. Confirmation of Urea in Soybean

We analyzed urea concentration using gas-liquid chromatography (GLC) together with the organic acids and carbohydrates after oximation and trimethylsilylation (TMS) derivatization (GL Science Inc., Tokyo, Japan). To confirm the presence of urea in xylem sap, the mass spectrum of TMS derivatives of the standard urea and the urea peak in the xylem sap were compared (Appendix A); similar fragmentation patterns were observed. We confirmed that the breakdown of allantoin and allantoate did not occur during derivatization and produced urea from ureides. We added an equal volume of the xylem sap and 5 mM of allantoin plus 5 mM of allantoate, and the urea concentrations were compared. The urea concentrations in the xylem sap (2.98 ± 0.49 mM) and xylem sap +5 mM allantoin plus 5 mM allantoate (2.91 ± 0.22 mM) were the same, and production of urea derived from ureides during derivatization and GC analysis was excluded. From these results, it was confirmed that urea was present in the xylem sap of the soybean plants.

### 2.9. Ratios of Urea-N/Ureide-N

The ratio of urea-N to ureide-N in each part of the control plants (Table 1) indicated that the ratio of the roots (0.398) was higher than that of the nodules (0.256), stems (0.047), and leaves (0.236). Similar trends were observed for the nitrate-, ammonium-, and urea-treated plants. These results suggest that the conversion of ureide to urea occurs actively in underground parts, especially in the roots. Of course, this conversion may also occur in stems and leaves.

## 3. Discussion

### 3.1. Characteristics of Nitrogen Transport Compounds in the Xylem

The nitrogen concentrations of N compounds in the xylem sap of soybean plants with various treatments are shown in Figure 9. The N concentrations were calculated from the molar concentration multiplied by the atomic weight of N (14) and the number of nitrogen atoms in one molecule. Ureides, allantoin, and allantoic acids, and Arg have four N atoms, His has three, and Asn, Gln, Lys, Trp, and urea have two atoms in one molecule. The other amino acids listed in Figure 9 have one nitrogen atom. Comparing N concentrations in the xylem sap, the changes of the dominant N compounds transported through the xylem in each treatment can be easily recognized.

In this section, a discussion of the characteristics of nitrogen transport compounds in the xylem is presented first, then, the possible urea formation in nodulated soybean is discussed; thirdly, the high accumulation of Asn and Gln is discussed; and finally, the effect of nitrogen treatment on sugar and organic acid concentrations is discussed.

The ureide concentration in the xylem sap was high, at 543 mgN/L; supplementation with Asn (66 mgN/L) and urea (61 mgN/L) in the control plants solely depended on the nitrogen fixation (Figure 9). On the other hand, the ureide concentration was relatively low at about 24.8 mgN/L, but nitrate accounts for over 50% of N (237 mgN/L) with Asn (106 mgN/L) in the nodulated plants with 5 mM nitrate treatment. Similar trends were shown in the non-nodulated plants with a continuous supply of 1 mM of nitrate. The severe decrease of ureide concentration in the xylem sap of the nodulated plants treated with nitrate may be related to the degree of repression of the nitrogen fixation activity with each compound. Similar compositions of N compounds were observed with the ammonium and urea treatments, in which ureide concentrations were about half of the control, and the concentrations of Asn and Gln were markedly increased. The concentrations of urea were almost the same as the control plants.

### 3.2. Urea Formation

Appreciable amounts of urea were detected in the xylem sap and all the organs (Figure 4B, Figure 9) for all treatments. Of the treatments, nitrate treatment severely decreased the urea concentration in the xylem sap to 0.62 mM, as compared with the ammonium (1.56 mM), urea (1.93 mM), and control (2.2 mM) treatments. Although urea was supplied in the urea treatment, the concentration of urea in the xylem sap was not higher than that of the control. On the other hand, the concentration of urea in the leaves, stems and roots was markedly increased by the ammonium treatment. Urea treatment also increased the concentration of urea in the roots, stems, and leaves, as compared with the control. Urea concentrations were lowest in all the organs in the non-nodulated plants.

In this experiment, the urea concentrations in leaves, stems, roots, and nodules were between 0.3–15 μmol/gDW (Figure 4B). Bohner et al. [34] reported that the urea concentration in young Arabidopsis leaves was about 5–10 μmol/gDW. The urea concentration in the radicle of 2-day-old soybean Williams 82 seedlings was about 2.1 μmol/gDW [35]. The urea concentrations in our experiment were roughly the same as in the previous reports.

Positive correlations were observed between the concentration of ureides and urea in the xylem sap (Figure 8A), nodules (Figure 8B), roots (Figure 8C), stems (Figure 8D), and leaves (Figure 8E), although these correlations were not observed between arginine and urea. This suggests that a major part of the urea in the xylem and soybean plants may originate from ureides and not from arginine, although this correlation does not provide direct evidence. The concentrations of ureide were much lower in the roots (3.5 μmol/gDW) than that in the nodules (18.3 μmol/gDW) in the control plants, and the ratios of urea-N/ureide-N (Table 1) were higher in the roots (0.398) than in the nodules (0.256), suggesting that ureide degradation may occur in the roots after transportation from the nodules. The same trends were observed in N-treated plants.

Most enzymatic or tracer researches on ureide degradation in soybean have been concentrated on the leaves, a sink organ of ureides [23,25,26,27,28,29,30], but some of the ureides may degrade in the roots and nodules. In nodules and roots, allantoate may be degraded by pathway A, as shown in Figure 1, releasing two urea molecules from one molecule of allantoate. Alternatively, urea may be released by the decay of S-ureidoglycine or S-ureidoglycolate through pathway B. Muñoz et al. purified and characterized the enzyme ureidoglycolate urea-lyase in chickpea [31] and French bean [32], which consists of six identical subunits and is dependent on manganese (Mn). They also observed the ureideglycolate degradation in various plant parts, including French bean roots [32]. The presence of this enzyme in soybean has not been reported, but it may contribute to urea formation from allantoate if present.

There is a possibility that some ureides such as allantoate, ureideglycine, or ureidoglycolate are chemically (non-enzymatically) hydrolyzed to urea under acidic conditions in the nodules, roots or transport processes through the xylem vessels. Further research is required to determine the origin of urea in soybean plants.

The alternative pathway of urea synthesis is the urea cycle, in which Arg is hydrolyzed with H_2_O to urea and ornithine by arginase [36]. In the urea cycle, ammonia is combined with HCO_3_^−^ and two adenosine triphosphate (ATP), and forms carbamoyl phosphate by carbamoyl phosphate synthase, and this binds with ornithine to produce citrulline by ornithine transcarbamoylase; then, citrulline is combined with Asp to make argininosuccinate by argininosuccinate synthetase, and then argininosuccinate forms arginine and fumarate by argininosuccinate lyase. A physiological role of this cycle may be to avoid the toxicity of ammonium accumulation and form non-toxic urea. In this study, severe ammonium accumulation was not observed in each part after 3 days of treatment of nitrogen compounds in all treatments including ammonium treatment (Figure 3B). Besides, the previous metabolome analysis 1 day after 5 mM nitrate treatment showed that the concentrations of the urea cycle intermediates arginine, ornithine, and citrulline in roots and nodules were one or two orders lower than Asn and Gln (Appendix A) and were increased a little by 1 day of 5 mM nitrate treatment in the roots. Therefore, the contribution of the urea cycle to forming urea maybe insignificant if present. Blume et al. [36] reported that Arabidopsis plants accumulated intermediate amino acids, under low CO_2_ conditions (100 ppm) in the urea cycle and suggested that the accumulation of ornithine and citrulline may be directly linked to an increase in ammonium assimilation that is stimulated by an increase in photorespiration under low CO_2_ conditions.

### 3.3. High Accumulation of Asn and Gln with Ammonium and Urea Treatments

The concentrations of Gln and Asn in the xylem sap were higher in the ammonium and urea treatments than in the control plants (Figure 9). The concentrations of Gln and Asn in the roots and stems were also very high in the ammonium and urea treatments. Using ^15^N-labeled urea, we observed that urea was actively absorbed from the culture solution and metabolized to amino acids and N compounds similar to ammonium, nitrate and glutamine [37]. The concentration of Asn in the nitrate treatment was higher than that of the control, although that of Gln was lower.

Similar metabolic patterns between urea and ammonium treatments may be due to the absorbed urea being rapidly hydrolyzed to ammonia by urease in the roots and then metabolized to Gln and Asn. Urea is a plant metabolite that is accumulated either from direct root uptake through high-affinity secondary active (DUR3) and passive (MIOs) urea transporters or from the catabolism of arginine by arginase [38] or by ureide degradation [28]. Urease is a unique nickel enzyme that hydrolyzes urea to ammonia and carbamate, and the carbamate rapidly decays to ammonia and carbon dioxide non-enzymatically. The soybean has two ureases [39]: a highly expressed embryo-specific urease encoded by the *Eu1* gene [38,40], and a ubiquitous urease found in all tissues encoded by the *Eu4* gene [38,41].

The concentration ratios of most of the major N compounds (Appendix A) and amino acids (Appendix A) to those in the control plants in the xylem sap and leaves, stems and roots tended to increase. However, nitrate treatment decreased the concentration ratios in the nodules by one fold. In a previous report of a transcriptome and metabolome analysis of soybean plants, comparing control and 5 mM nitrate treatments for 1 day, the Asn concentration and the expression of asparagine synthetase (AS) in the nodules were exceptionally high in nitrate-treated plants [42]. This phenomenon was not observed in this experiment after 3 days of treatments. The transient promotion of Asn synthesis might occur in the nodules after 1 day of nitrate supply. Huber and Streeter [43] reported that glutamine-dependent asparagine synthetase catalyzes the amidation of aspartate to Asn in the cytosolic fraction of the infected zone of soybean nodules. Minamisawa et al. [44] estimated the Asn and ureide pools in soybean nodules and found that the amount of fixed N in Asn-N was about one-fifth of ureide-N. Ohtake et al. [45] reported that the major form of amino acids in the xylem sap of soybean plants was Asn in a field undergoing various fertilizer treatments.

### 3.4. Decrease in Sugars and Organic Acids Concentrations by N Treatments

The concentrations of fructose (0.6 mM) and glucose (0.5 mM) in the xylem sap of the control plants were higher than that of sucrose (0.3 mM) and myo-inositol (0.24 mM). For the nitrogen treatments, the sucrose concentration in the xylem sap was decreased, although the concentrations of glucose and fructose decreased only slightly (Appendix A). It is well known that sugars, especially sucrose, are transported from the shoot to the roots and nodules through the phloem, and little research has focused on the transport of sugars in the xylem. Some parts of the sugars in the xylem sap may probably be derived from the root apoplast or transferred from the phloem.

The concentration ratios of organic acids in the nitrogen treatments to the control (Appendix A) tended to be lower by one fold. This may be due to the higher consumption of carbohydrates and organic acids to assimilate N supplied from the culture solution. The concentrations of organic acids in the xylem sap of control plants were higher for malate (2.5 mM) and citrate (0.9 mM) than α-KG (0.24 mM), succinate (0.18 mM) and fumarate (0.12 mM) in this sequence. Nitrate treatments decreased the organic acid concentrations in the xylem sap, especially for malate. Malate is a major energy and carbon source supplied to bacteroids to support nitrogen fixation [46]. The decrease in malate concentration in the xylem sap may be related to the stimulation of malate consumption in the roots and nodules. Nitrate supply most severely reduced the concentrations of all organic acids in the xylem sap, partly due to the fact that nitrate acts as a counter anion for transporting cations, such as K^+^, Ca^2+^, and Mg^2+^ replaced for organic acids. The decrease in organic acids in the xylem sap under the nitrate treatment may decrease metal transport, such as for Fe, because organic acids act as chelating compounds for metals in xylem transport [47].

## 4. Materials and Methods

### 4.1. Plant Material and Growth Conditions

About 50 seeds of the soybean (*Glycine max* (L.) Merr. cv. Williams) were inoculated with 50 mL of a suspension of *Bradyrhizobium diazoefficiens* (strain USDA 110) from a yeast mannitol agar slant in a test tube and sown in a vermiculite bed. After 5 DAP, each seedling was transplanted to a glass bottle with 800 mL of N-free nutrient solution [48]. The culture solution was continuously aerated by an air pump and changed three times a week. Plants were cultivated in a climate chamber (MLR-350; Sanyo, Osaka, Japan), at 28/18 °C of for the day/night temperature, 55% relative humidity, 228 μmol m^−2^ s^−1^ photosynthetic photon flux density (PPFD), and 16/8 h day/night photoperiod). Non-nodulated plants were sown without inoculation of rhizobium and cultured with a culture solution containing 1 mM NaNO_3_ until 23 DAP.

### 4.2. Application of Nitrogen Compounds and Plant Sampling

Culture solutions containing 5 mM N sources (5 mM NaNO_3_, 5 mM NH_4_Cl, or 2.5 mM urea) were supplied to the nodulated plants for 3 d from 20 to 23 DAP. We selected 3 days of treatment because the effects of nitrogen application on nodule growth and nitrogen fixation activity can be observed after a few days of treatment [49]. We used NaNO_3_ instead of KNO_3_ to avoid K concentration changes among the treatments. The solution was renewed every day during the treatment period. Control plants were grown with an N-free culture solution throughout the experimental period. At 23 DAP, xylem sap was collected for 1 h from the basal part of the stem by cutting the shoots [49]. The shoots and roots were then dried using a freeze drier (VD-800F, TAITEC, Saitama, Japan) and separated into the leaves, stems plus petioles, roots, and nodules, and the dry weight was measured. These parts were ground into a fine powder with a vibration mill (CMT, Tokyo, Japan), and approximately 50 mg of dry matter was extracted with 1mL of 80% (v/v) ethanol and kept at 60 °C for 15 min [50] to promote protein denaturation and inactivation of enzyme activity such as the asparaginase activity. After vortexing for 10 min, the plastic tube was centrifuged at 14,000 rpm at 4 °C for 15 min, and the extract was separated. The residue was washed twice with 1 mL of 80% ethanol following the aforementioned procedure and evaporated to dryness. Then, 200 μL of water was added to the dry residue and dissolved by vortexing for 10 min and centrifuged for 15 min at 4 °C at 14,000 rpm. By analyzing the capillary electrophoresis (7100, Agilent Technologies, Inc., California, USA), with a fused silica tube (inner diameter, 50 μm; length 104 cm) using a commercial buffer (α-AFQ 109, Agilent Technologies, Inc., Santa Clara, CA, USA) with an applied voltage of −25 kV, we confirmed that the degradation of allantoin to allantoate and allantoate to glyoxylate did not occur at all during incubation with 80% ethanol at 60 °C for 15 min

The experiment was replicated using four independent plants, and all the treated plants were cultivated simultaneously under the same conditions. Data in figures were presented as the average values with standard error, and the statistical analysis was performed using Tukey’s test.

### 4.3. Analysis of Free Amino Acids and Ammonium

The free amino acids and ammonium in the extracts of roots, nodules, stems, and leaves of the plants and xylem sap were derivatized with 6-aminoquinolyl-*N*-hydroxysuccinimidyl-carbamate (AQC) reagent (AccQ·Flour, Waters, Milford, MA, USA), and analyzed using a Waters Aquity UPLC system equipped with a Waters AccQ Tag Ultra column (2.1 mm × 100 mm, particle size 1.7 μm; Waters, Milford, MA, USA). The column temperature was 60 °C and the sample temperature was 20 °C. The wavelength for the detection of AQC-amino acid derivatives was at 260 nm using an Aquity UPLC TUV Detector. Elution was performed using a gradient of Waters AccQTag Ultra Eluent A and Eluent B with a flow rate of 0.7 mL/min [51]. For the internal standard, 2.5 mM 2-aminoisobutylic acid (AABA) was used. For the calibration of amino acids, 200 μM of a standard amino acid mixture was used (18 amino acids by Amino Acid Standard H, Thermo Fisher Scientific Ltd, Waltham, MA, USA. with added GABA, Trp, Asn, Gln, and AABA).

### 4.4. Analysis of Anion

The anion concentrations in the plant organ extracts and xylem sap were determined by capillary electrophoresis (7100, Agilent Technologies, Inc., Santa Clara, CA, USA) using a fused silica tube (inner diameter (id); 75 μm, 72 cm long) and the commercial buffer α-AFQ108, (Agilent Technologies, Inc.) with an applied voltage of −15 kV. Peaks were detected with a signal wavelength of 400 nm and a reference wavelength of 265 nm.

### 4.5. Analysis of Urea, Sugars, and Organic Acids

The concentrations of urea, sugars, and organic acids in the extract and xylem sap were determined by gas chromatography equipped with a flame ionization detector (FID) (GC2014, Shimadzu Corporation, Japan) using a capillary column with id; 0.25 μm coated with 5% diphenyl and 95% dimethylpolysilarylene (InertCap 5MS/NP; GL Sciences Inc., Tokyo, Japan) and helium gas was used for the mobile phase. Ten microliters of extract or standard solution plus 2 μL of 2 mM ribitol (internal standard) were placed in a 1.5 mL plastic tube, and dried in vacuo overnight. One hundred microliters of 20 mg/mL methoxyamine HCl in pyridine was added to the tube and was agitated for 90 min at room temperature around 25 °C. Then, 100 μL of N-methyl-N-TMS-trifluoroacetamide (MSTFA) was added for the oximation of urea, sugars, and organic acids to TMS derivatives. The column temperature was set at 80 °C for 2 min, then increased to 330 °C at a rate of 15 °C/min. For the calibration, a mixture of 10 mM of urea, sucrose, glucose, fructose, myo-inositol, citrate, α-ketoglutaric acid, succinate, fumarate, malate, and malonate was used.

### 4.6. Analysis of Urea by Gas Chromatograph/Mass Spectrum

The presence of urea in the plant extract and xylem sap was confirmed by analyzing the mass spectrum of the TMS derivatives of urea with a GC/MS (Gas Chromatograph/Mass Spectrum M-9000 GC/3DQMS, Hitachi, Tokyo, Japan) fitted with a capillary column (InertCap 5MS/NP; Shimadzu, Kyoto, Japan).

### 4.7. Analysis of Ureides

The quantitative analysis of ureides, which are the sum of allantoin and allantoate, was carried out using the modified micro-scale method of Young and Conway [49]. Ten microliters of the sample solution were added to a 1.5 mL plastic tube, and 300 μL of 83 mM NaOH solution was mixed. This was heated at 100 °C for 8 min to hydrolyze the allantoin to allantoate. After cooling the solution, 100 μL of ice-cold acidic phenylhydrazine hydrochloride solution, and heated at 100 °C for 2 min to hydrolyze allantoate to urea and glyoxylate. Then, 125 μL of ice-cold potassium ferricyanide solution was mixed and incubated in an ice-water bath for 30 min. The absorption at 520 nm of 200 μL of the reaction mixture in each well of a 96-well flat-bottom microplate was measured using a microplate reader (Multiskan Spectrum, Thermo Electron, Vantaa, Finland) after 30 min of incubation in an iced water bath.

## 5. Conclusions

Nodulated soybean plants were supplied with 5 mM N of nitrate, ammonium, or urea for 3 d, and the concentrations of ureides, urea, amino acids, anions, sugars, and organic acids in the xylem sap and each organ were analyzed. Considerable amounts of urea were universally present in the xylem sap and all organs with all treatments. A positive correlation was observed between the concentrations of ureides and urea in the xylem sap as well as in the leaves and roots, although correlations were not observed between urea and arginine, except for in leaves. The concentrations of ureides were lower in the roots than that in the nodules and the ratios of urea-N/ureide-N were higher in the roots than that in the nodules, suggesting that ureide degradation may occur in the roots after ureides are transported from nodules. These results suggest that a major part of urea in the nodulated soybean plant may originate from ureide degradation, and urea production may occur in roots. The experiment here only analyzed one-time point, 3 days after the application of N compounds, so it is necessary to investigate the time-course changes of metabolite concentrations after the addition of treatments. Further studies are required to confirm ureide degradation and urea production in soybean roots and other organs.

## Figures and Tables

**Figure 1 ijms-22-04573-f001:**
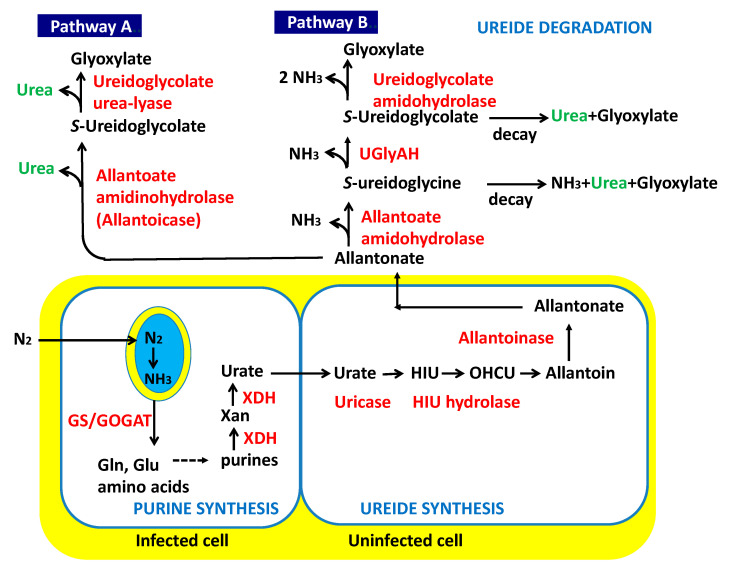
A model of ureide synthesis in the nodules and ureide degradation in soybean. GS: glutamine synthetase; GOGAT: glutamate synthase; XDH: xanthine dehydrogenase; Xan: xanthine; HIU: hydroxyisourate; OHCU: 2-oxo-4-hydroxy-4-carboxy-5-ureidoimidazoline; UGlyAH: ureidoglycine aminohydrolase.

**Figure 2 ijms-22-04573-f002:**
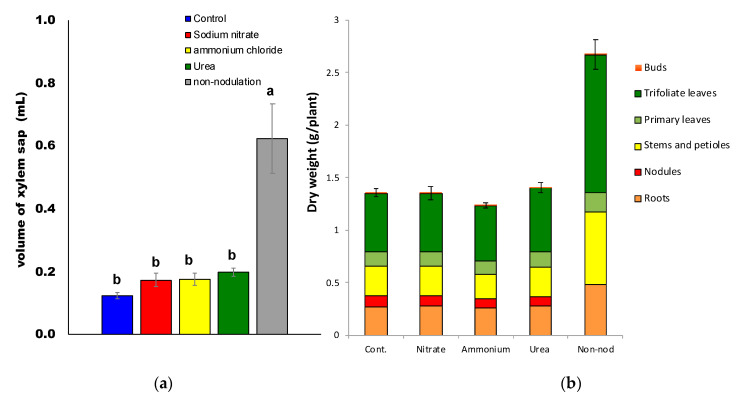
The volume of xylem sap (**a**) and dry weight (**b**) of soybeans after 3 d of N treatment 23 DAP. Different letters above the bars indicate significant differences (*p* < 0.05) by Tukey’s test. Error bars indicate standard error (SE) from the four plants.

**Figure 3 ijms-22-04573-f003:**
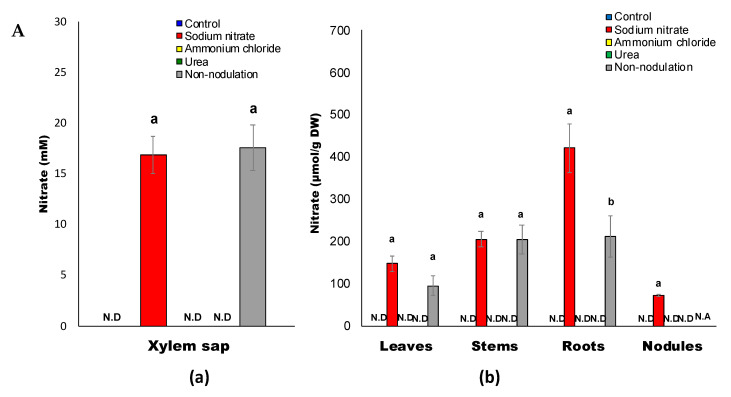
(**A**) Nitrate concentrations in the xylem sap (**a**) and in each part (**b**) of the soybean plant. (**B**) Ammonium concentrations in the xylem sap (**a**) and in each part (**b**) of the soybean plant. N.A: not available; N.D: not detected.

**Figure 4 ijms-22-04573-f004:**
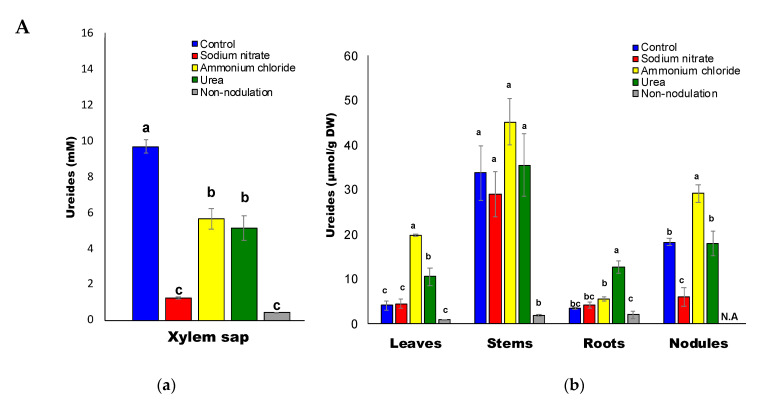
(**A**) Ureide concentrations in the xylem sap (**a**) and in each part (**b**) of the soybean plant. (**B**) Urea concentrations in the xylem sap (**a**) and in each part (**b**) of the soybean plant.

**Figure 5 ijms-22-04573-f005:**
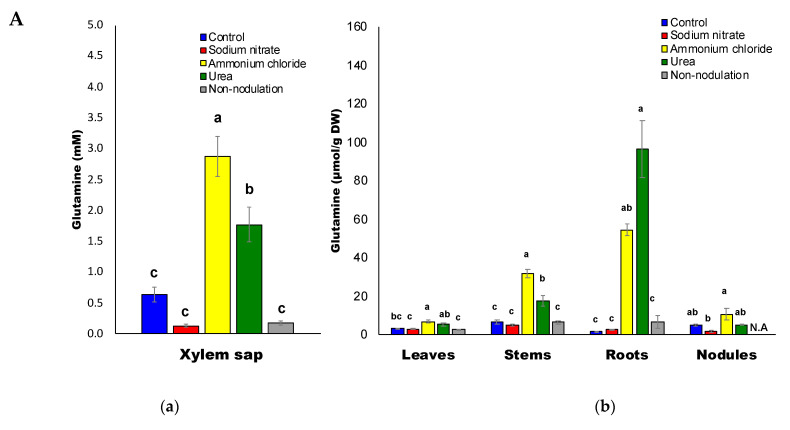
(**A**) Glutamine concentrations in the xylem sap (**a**) and each part (**b**) of the soybean plant. (**B**) Asparagine concentrations in the xylem sap (**a**) and in each part (**b**) of the soybean plant. (**C**) Glutamate concentrations in the xylem sap (**a**) and in each part (**b**) of the soybean plant. (**D**) Aspartate concentrations in the xylem sap (**a**) and in each part (**b**) of the soybean plant. (**E**) Arginine concentrations in the xylem sap (**a**) and in each part (**b**) of the soybean plant. N.A: not available.

**Figure 6 ijms-22-04573-f006:**
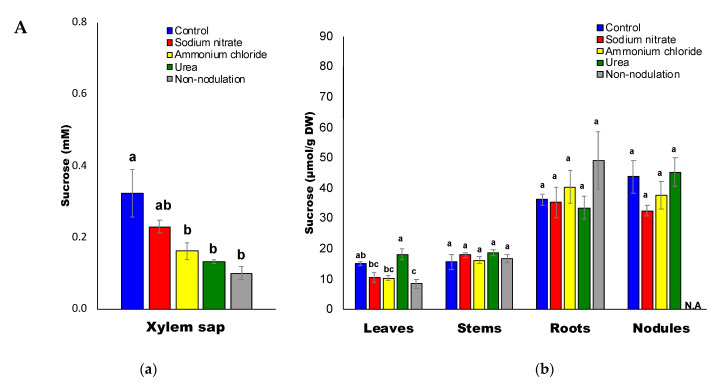
(**A**) Sucrose concentrations in the xylem sap (**a**) and in each part (**b**) of the soybean plant. (**B**) Glucose concentrations in the xylem sap (**a**) and in each part (**b**) of the soybean plant. (**C**) Fructose concentrations in the xylem sap (**a**) and in each part (**b**) of the soybean plant. (**D**) Myo-inositol concentrations in the xylem sap (**a**) and in each part (**b**) of the soybean plant. N.A: not available.

**Figure 7 ijms-22-04573-f007:**
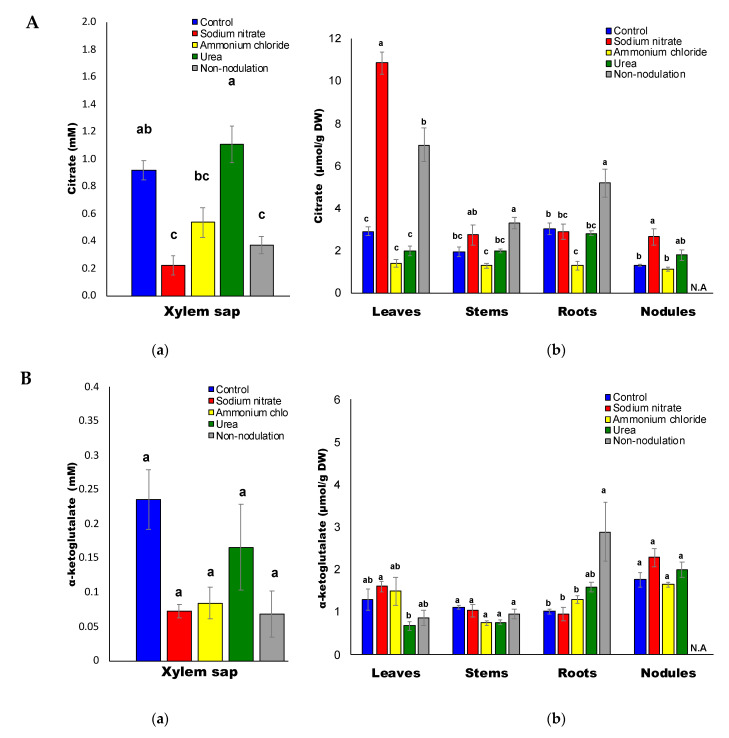
(**A**) Citrate concentrations in the xylem sap (**a**) and in each part (**b**) of the soybean plant. **(B).** α-Ketoglutarate concentrations in the xylem sap (**a**) and in each part (**b**) of the soybean plant. (**C**) Succinate concentrations in the xylem sap (**a**) and in each part (**b**) of the soybean plant. (**D**) Fumarate concentrations in the xylem sap (**a**) and in each part (**b**) of the soybean plant. (**E**) Malate concentrations in the xylem sap (**a**) and in each part (**b**) of the soybean plant. (**F**) Malonate concentrations in the xylem sap (**a**) and in each part (**b**) of the soybean plant. N.A: not available.

**Figure 8 ijms-22-04573-f008:**
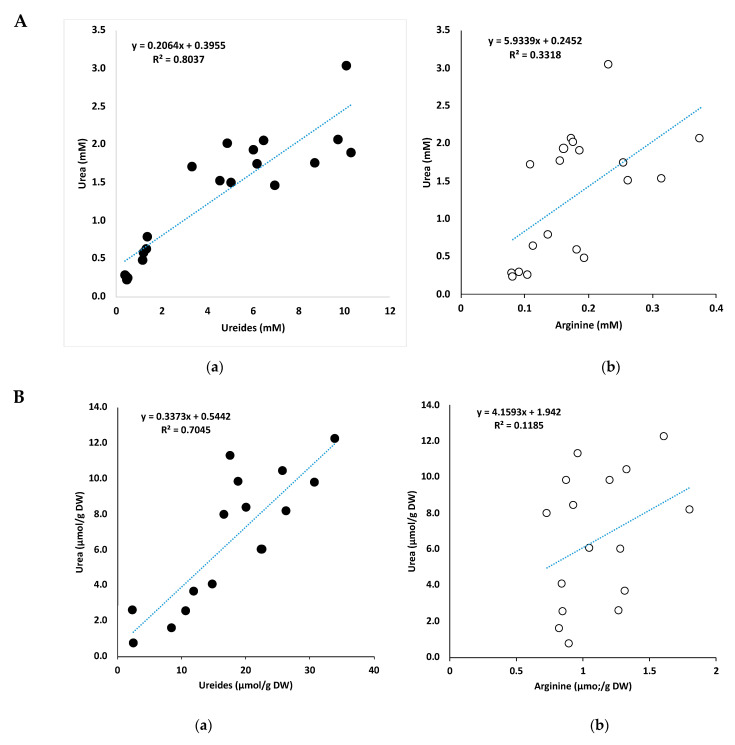
(**A**) Correlations between the concentrations of ureides and urea (**a**) and arginine and urea (**b**) in the soybean xylem sap. (**B**) Correlations between the concentrations of ureides and urea (**a**) and arginine and urea (**b**) in the soybean nodules. (**C**) Correlations between the concentrations of ureides and urea (**a**) and arginine and urea (**b**) in the soybean roots. (**D**) Correlations between the concentrations of ureides and urea (**a**) and arginine and urea (**b**) in the soybean stems. (**E**) Correlations between the concentrations of ureides and urea (**a**) and arginine and urea (**b**) in the soybean leaves.

**Figure 9 ijms-22-04573-f009:**
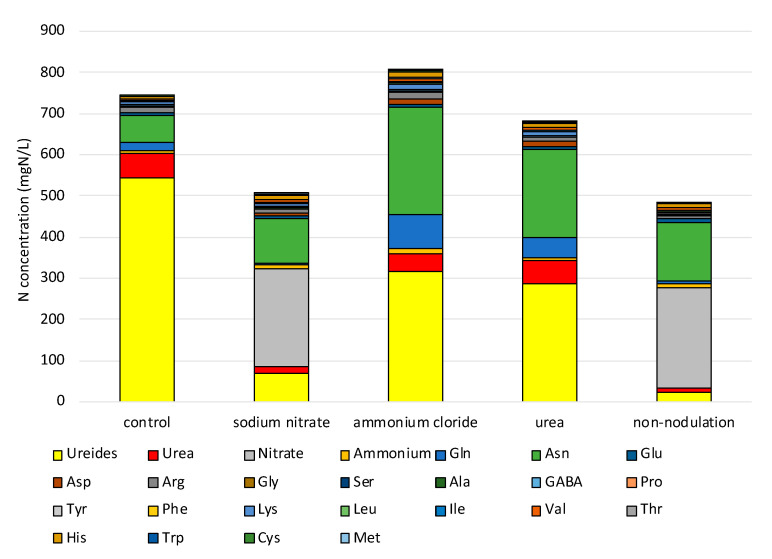
Nitrogen concentrations of N compounds in xylem sap of soybean.

**Table 1 ijms-22-04573-t001:** Ratios of urea-N/ureide-N in the xylem sap and in each part of the soybean plants.

Treatment	Xylem Sap	Nodules	Roots	Stems	Leaves
Control	0.113	0.256	0.398	0.047	0.236
Sodium nitrate	0.246	0.154	0.458	0.07	0.212
Ammonium chloride	0.138	0.174	0.613	0.165	0.205
Urea	0.187	0.137	0.464	0.13	0.194
Non-nodulation	0.292		0.158	0.11	0.164

## Data Availability

Data in Appendix A was based on the metabolome analysis data in Appendix A of the reference [42]: https://www.mdpi.com/2223-7747/7/2/32#supplementary (accessed on 1 January 2021).

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
