# Peer review of "Application of Nitrate, Ammonium, or Urea Changes the Concentrations of Ureides, Urea, Amino Acids and Other Metabolites in Xylem Sap and in the Organs of Soybean Plants (*Glycine max* (L.) Merr.)"

_ijms, 2021, doi:10.3390/ijms22094573_

Round 1

Reviewer 1 Report

The authors addressed correctly the changes that I suggested. I have no further comments.

Author Response

Dear reviewer 1

Thank you very much for reviewing our manuscript and gave us proper comments and excellent suggestions. The manuscript has been much improved compared with the original one. I changed a little by the comments of reviewer 2.

Reviewer 2 Report

This paper is a resubmission of a previous paper that describes the effects on different metabolite levels of nodulated soybean plants of supplementing them with different forms of nitrogen (specifically nitrate, ammonium and urea). Since I met this paper for the first time, the several review rounds have improved it a bit, but I think the discussion did not improve enough, even though the reviewers insisted on it several times.

I think the title I suggested in the third and last report (Application of nitrate, ammonium or urea changes the concentrations of ureides, urea, amino acids and other metabolites in xylem sap and in the organs or soybean plants (Glycine max (L.) Merr.) grammatically is more correct than the one the authors use. Any case, if they prefer theirs, it should grammatically be “Application of nitrate, ammonium, or urea changes ureide, urea, amino acid and other metabolite concentrations in xylem sap and in the organs of soybean plants (Glycine max (L.) Merr.)”, without several “s”. The same could be said for a “ureides” in the line 22 in the summary.

The introduction section has now an appropriate length and, in general terms, is correct.

The result section is OK as well, even though, by the end of it, a table is introduced (line 487) that apparently is missing. Later I realised the authors moved this table to the supplementary information. Please, correct this.

The discussion section, under my point of view, is not enough worked. Again they just changed a bit this part. The introduction of figure 9 is a wise choice and I think the section could start talking about it and not directly talking about this or that concentration. The discussion is based on 4 aspects, corresponding to its 4 sections, but nowhere there is an explanation why these aspects are the most relevant in this study. This could be another good way to start, before the introduction of figure 9, which is useful for 3 out of the 4 sections. Apart from that, the new introduced paragraphs are interesting and agree with which the rest of the discussion should be; however, their English grammar should be re-revised, because there are some mistakes, like the misuse of hyphens (for instance urea-treatment, ammonium treatment, urea-cycle, etc.) or of “highest” (which in any case should be “the highest”) that should be “higher” when it is in a comparison. In general, another good revision of English could be very beneficial for the paper.

M&M is more or less OK.

As I said before the conclusion section “is mainly a summary of the obtained results that could be good for the actual summary”. I think better conclusions could be withdrawn from the results apart from “a major part of urea in the nodulated soybean plant may originate from ureide degradation, and urea production may occur in roots”.

Author Response

Thank you very much for your repeated review of our manuscript and gave us excellent suggestions to improve our manuscript.

1) This paper is a resubmission of a previous paper that describes the effects on different metabolite levels of nodulated soybean plants of supplementing them with different forms of nitrogen (specifically nitrate, ammonium and urea). Since I met this paper for the first time, the several review rounds have improved it a bit, but I think the discussion did not improve enough, even though the reviewers insisted on it several times.

Answer: Thank you very much for reviewing our paper many times, and give us proper comments and excellent suggestions. I am sorry that our revision was not enough to improve to be published. Based on your comments we seriously revised the discussion parts.

2)  I think the title I suggested in the third and last report (Application of nitrate, ammonium or urea changes the concentrations of ureides, urea, amino acids and other metabolites in xylem sap and in the organs or soybean plants (Glycine max (L.) Merr.) grammatically is more correct than the one the authors use. Any case, if they prefer theirs, it should grammatically be “Application of nitrate, ammonium, or urea changes ureide, urea, amino acid and other metabolite concentrations in xylem sap and in the organs of soybean plants (Glycine max (L.) Merr.)”, without several “s”. The same could be said for a “ureides” in the line 22 in the summary.

Answer: Thank you very much for the valuable suggestion for the title. I changed the title as your suggestion and English editor by MIPD.

3)  The introduction section has now an appropriate length and, in general terms, is correct.

Answer: Thank you very much. I am sorry that the previous introduction was too redundant.

4)  The result section is OK as well, even though, by the end of it, a table is introduced (line 487) that apparently is missing. Later I realised the authors moved this table to the supplementary information. Please, correct this.

Answer: I revised the Table 1 in the text from supplementary information.

5)  The discussion section, under my point of view, is not enough worked. Again they just changed a bit this part. The introduction of figure 9 is a wise choice and I think the section could start talking about it and not directly talking about this or that concentration. The discussion is based on 4 aspects, corresponding to its 4 sections, but nowhere there is an explanation why these aspects are the most relevant in this study. This could be another good way to start, before the introduction of figure 9, which is useful for 3 out of the 4 sections.

Answer: I greatly appreciate your good suggestion. I added two paragraphs from line 501-511 to introduce Figure 9 instead of sudden explanation of the data.

6)  Apart from that, the new introduced paragraphs are interesting and agree with which the rest of the discussion should be; however, their English grammar should be re-revised, because there are some mistakes, like the misuse of hyphens (for instance urea-treatment, ammonium treatment, urea-cycle, etc.) or of “highest” (which in any case should be “the highest”) that should be “higher” when it is in a comparison. In general, another good revision of English could be very beneficial for the paper.

Answer: I am very sorry that our English is not good. Although I asked English edition once, I asked MDPI English edition again.

7)  M&M is more or less OK.

Answer: Thank you very much.

8)  As I said before the conclusion section “is mainly a summary of the obtained results that could be good for the actual summary”. I think better conclusions could be withdrawn from the results apart from “a major part of urea in the nodulated soybean plant may originate from ureide degradation, and urea production may occur in roots”.

Answer: As your comment, I deleted most part of conclusion except for “a major part of urea in the nodulated soybean plant may originate from ureide degradation, and urea production may occur in roots”. The conclusion was much better.

Round 2

Reviewer 2 Report

I think the paper now is improved, but the title should be "Application of nitrate, ammonium, or urea changes the concentrations of ureides, urea, amino acids and other metabolites in xylem sap and in the organs of soybean plants (Glycine max (L.) Merr.)"

This manuscript is a resubmission of an earlier submission. The following is a list of the peer review reports and author responses from that submission.

Round 1

Reviewer 1 Report

The manuscript no. ijms-992789 reports results on the investigation of metabolites of nodulated soybean plants supplied with different forms of nitrogen. Nodulation of plants was induced by Bradyrhizobium diazoefficiens. After 20 days of planting the nodulated plants were treated with 5 mM N nitrate, ammonium, and urea for 3 d, and the concentrations of metabolites in the xylem sap and in each organ were analyzed. A possible urea formation by ureide degradation has been proposed. 

MAJOR COMMENTS

I think that the subject of the manuscript is interesting and could add knowledge to the field. However, the quality of presentation is low and it requires an extensive review.

INTRODUCTION

Authors must revise this section thoroughly. This section should be much briefer, highlighting better why the study is important. The working hypothesis is missing.

RESULTS 

The presentation of the results is almost correct. However, the supplementary figures S1, S2, S3, S4, S5 are completely missing. Authors should also ameliorate figures captions, presenting the details of statistical analysis carried out and case letters meaning.

Figure 3b - are the error bars presented the correct ones? 

DISCUSSION

Authors did not discuss the results properly. There is limited interpretation in perspective of the working hypothesis and of literature.

L628-632 - this paragraph and Table 1 should be moved in the results section.

MATERIAL AND METHODS

Please provide more details about:

  • clear experimental design with the number of plants per experimental condition.
  • Details about Bradyrhizobium diazoefficiens inoculation (i.e. inoculum density, inoculum:seeds ratio; method and time of expsure).
  • L671 - cite method or describe the procedure followed for capillary electrophoresis.
  • L681 - Add Detector type details. 
  • L695 - Add Column details.
  • L716 - Add volume measured and type of microplate used.

CONCLUSIONS

Future research directions should be mentioned.

MINOR COMMENTS

  • L620 - The item "(Rellan-Alvarez et al. 2011)" should be substituted with "[57]".
  • L666, L670 - Maybe authors meant "vortexing" and not "vibration". 
  • L674, I suggest providing the details of all statistical analyses carried out in a separated section. 
  • Abbreviations should be defined in parentheses the first time they appear in the abstract, main text, and in figure or table captions and used consistently thereafter.
  • References: Journal style has not been followed properly. Items should be revised for both their format (e.g. doi presentation, volume no. in italics) and contents (missing details in item no. 57 or 59 for example).

Author Response

Thank you very much for your kind and constructive comments and suggestions for our manuscript.

First of all, I am very sorry that all supplementary figures are missing in the text. I submitted them as a separate file, but I should have attached them in the text.

I attached all the supplementary figures in the text.

I revised following your advice as below, and the revised parts were heightened by a yellow background.

The manuscript no. ijms-992789 reports results on the investigation of metabolites of nodulated soybean plants supplied with different forms of nitrogen. Nodulation of plants was induced by Bradyrhizobium diazoefficiens. After 20 days of planting the nodulated plants were treated with 5 mM N nitrate, ammonium, and urea for 3 d, and the concentrations of metabolites in the xylem sap and in each organ were analyzed. A possible urea formation by ureide degradation has been proposed. "

MAJOR COMMENTS

I think that the subject of the manuscript is interesting and could add knowledge to the field. However, the quality of presentation is low and it requires an extensive review.

INTRODUCTION

Authors must revise this section thoroughly. This section should be much briefer, highlighting better why the study is important. The working hypothesis is missing.

I deleted the last paragraph of the introduction and the initial part was moved to the results section according to the comment of Reviewer 2.

I added the working hypothesis at the first part of the introduction.

L30-34 (revised manuscript)

In the present study, we investigated the effect of the application of combined nitrogen (N), nitrate, ammonium, and urea, on the metabolite concentrations of the xylem sap and the plant parts of nodulated and non-nodulated soybean, whether different N forms may affect the metabolism and transport of various metabolites either from nitrogen fixation by nodules or the N absorption from the roots.

RESULTS 

The presentation of the results is almost correct. However, the supplementary figures S1, S2, S3, S4, S5 are completely missing. Authors should also ameliorate figures captions, presenting the details of statistical analysis carried out and case letters meaning.

I am very sorry that all supplementary figures are missing in the text. I submitted them as a separate file, but I should have attached them in the text.

I attached all the four supplementary figures in the text.

Figure 3b - are the error bars presented the correct ones? 

Thank you for the suggestion. I confirmed the error bars in Figure 3 were correct.

DISCUSSION

Authors did not discuss the results properly. There is limited interpretation in perspective of the working hypothesis and of literature.

Thank you very much for your advice. I added the discussion part.

L628-632 - this paragraph and Table 1 should be moved to the results section.

I agree with your suggestion. I moved Table 1 and the explanation from discussion to the end of the result section.

MATERIAL AND METHODS

Please provide more details about:

Thank you for your advice. I revised as follows:

  • clear experimental design with the number of plants per experimental condition.
  • L736-738: The experiment was replicated from four plants, and all the treated-plants were cultivated simultaneously under the same conditions. Data in figures were presented as the average values with standard error, and the statistical analysis was performed using Tukey’s test.
  •  
  • Details about Bradyrhizobium diazoefficiens inoculation (i.e. inoculum density, inoculum:seeds ratio; method and time of exposure).
  • L706-708: About 50 seeds of Soybean (Glycine max (L.) Merr. cv. Williams) were inoculated with 50 mL of a suspension of Bradyrhizobium diazoefficiens (strain USDA 110) from a YM-agar slant in a test tube and sown in a vermiculite bed.

  • L671 - cite method or describe the procedure followed for capillary electrophoresis.

I added the detailed procedures for capillary electrophoresis.

L731-735: By analyzing the capillary electrophoresis (7100, Agilent Technologies, Inc., California, USA), with a fused silica tube (inner diameter, 50 μm; length 104 cm) using a commercial buffer (a-AFQ 109, Agilent Technologies, Inc.) with an applied voltage of -25 kV, we confirmed that the degradations of allantoin to allantoate and allantoate to glyoxylate did not occur at all during incubation with 80% ethanol at 60 oC for 15 min

  • L681 - Add Detector type details. 

L745-746: The wavelength for the detection of AQC-amino acid derivatives was at 260 nm using an Aquity UPLC TUV Detector.

  • L695 - Add Column details.

L 758-761: The concentrations of urea, sugars, and organic acids in the extract and xylem sap were determined by gas chromatography equipped with an FID detector (GC2014, Shimadzu Corporation, Japan) using a capillary column with id; 0.25 μm coated with 5% diphenyl - 95% dimethylpolysilarylene (InertCap 5MS/NP; GL Sciences Inc., Tokyo, Japan) and helium gas was used for the mobile phase.

  • L716 - Add volume measured and type of microplate used.

L782-785: The absorption at 520 nm of 200 mL of the reaction mixture in each hole of a 96-hole flat bottom microplate was measured using a microplate reader

CONCLUSIONS

Future research directions should be mentioned.

I added future research at the bottom of the conclusion.

L799-802: The experiment here was only analyzed one time after the application of N compounds, so it is necessary to investigate the time-course changes of metabolite concentrations after the addition. Further studies are required to confirm the ureide degradation and urea production in soybean roots and nodules.

MINOR COMMENTS

  • L620 - The item "(Rellan-Alvarez et al. 2011)" should be substituted with "[57]". I changed as your comment.
  • L666, L670 - Maybe authors meant "vortexing" and not "vibration". I changed as your comment.
  • L674, I suggest providing the details of all statistical analyses carried out in a separated section. 
  • Abbreviations should be defined in parentheses the first time they appear in the abstract, main text, and in figure or table captions and used consistently thereafter.

I deleted the abbreviation list, instead, all the abbreviations are defined in parentheses at the first time they appear as your suggestion.

  • References: Journal style has not been followed properly. Items should be revised for both their format (e.g. doi presentation, volume no. in italics) and contents (missing details in item no. 57 or 59 for example).

Thank you for your comments. I changed volume no. in italics.

Doi presentation seems to be changed by the editor such as [CrossRef] or [Pubmed].

Thank you very much for spending your valuable time revising our manuscript.

We appreciate your excellent comments for us.

Reviewer 2 Report

This paper describes the effects on different metabolite levels of nodulated soybean plants of supplementing them with different forms of nitrogen (specifically nitrate, ammonium and urea). The main conclusion they seem to obtain is that ureide degradation in the different organs yields urea, and specially they remark that, for the first time, this is described in underground tissues. In general terms the paper is very descriptive (as it is the title, which I think it should be more conclusive), and from an experiment (I do not see clearly the number of used biological replicates) the authors measure all the different metabolites related with nitrogen, carbon, phosphorus, etc. However, from all the measurements carried out, only a few conclusions are withdrawn. Perhaps a deeper analysis of the same results could help to get more conclusions.

The English language in general terms is correct, even though there are some mistakes that will be detailed below.

The introduction gives enough background, but in some aspects it is too long, and in some others some references are missing. For example, as the main conclusion is related to ureide degradation to urea, I missed the references about two enzymes that were purified from legumes and that were characterised initially as ureidoglycolate urea-lyases (Muñoz et al, 2001 and 2006), even though they seem to catalyse another reaction that eventually could produce urea (Muñoz et al, 2010). The final paragraph is a bit weird because generally is summarizing and introducing what the authors did; however, the description that they did could be more the beginning of the Results section. I think that only the sentences between lines 160 and 163 in this paragraph should be kept in the introduction.

The way of describing the results is very repetitive with “Figure X shows…”. I think the authors should be a bit more imaginative about the writing in order to not to be boring in the description. Perhaps I missed it, but I could not find the supplementary figures of results and the heatmap in the figure 8A (line 589) referred in the discussion. In the figure 4A, some letters for the Tukey’s test are missing.

The Discussion section is very descriptive as well, and a good part of it is a repetition of the results. There are just a few references to other papers related to this. I think it could be richer if the authors had compared their results with those from some other papers (either similar or dissimilar). Furthermore, the figures referred in this section are not corresponding with the actual ones.

The Materials and methods section is, in general, well described; however, it would be good to describe the way that the fine powder was obtained (line 664). As I noted before, I do not know if the replicates the authors talk about (line 674) are actual biological replicates (plants grown independently four times), or four samples from a unique experiment.

The Conclusions section is mainly a summary of the obtained results that could be good for the actual summary. Only the final sentence could be a conclusion.

The Abbreviations section is too long. I think it is not necessary to put the three-letter code for amino acids or some others that are in the text, as DAP.

Minor points

- I think that the authors may have something to acknowledge to someone. Please, put it in the corresponding section.

- In the Bibliography section, there are some references that do not follow the standard pattern followed in most of them (some with pp. for the pages and some others without it). In number 6 the year is missing.

- There are some references in the body text that are in a different format from numbers (line 620).

- Line 445: Instead of using xX, I think it would have been better to use X-fold all over the paper.

- Please, have a look and correct some grammar mistakes that are scattered in the body text.

Author Response

Thank you very much for your kind and constructive comments and suggestions for our manuscript.

First of all, I am very sorry that all supplementary figures are missing in the text. I submitted them as a separate file, but I should have attached them in the text.

I attached all the supplementary figures in the text.

I revised following your advice as below, and the revised parts were heightened by a yellow background.

This paper describes the effects on different metabolite levels of nodulated soybean plants of supplementing them with different forms of nitrogen (specifically nitrate, ammonium and urea). The main conclusion they seem to obtain is that ureide degradation in the different organs yields urea, and specially they remark that, for the first time, this is described in underground tissues. In general terms the paper is very descriptive (as it is the title, which I think it should be more conclusive), and from an experiment (I do not see clearly the number of used biological replicates) the authors measure all the different metabolites related with nitrogen, carbon, phosphorus, etc. However, from all the measurements carried out, only a few conclusions are withdrawn. Perhaps a deeper analysis of the same results could help to get more conclusions.

I changed the title “Changes in metabolite concentrations in the xylem sap and organs of soybean plants (Glycine max (L.) Merr.) supplied with chemical nitrogen, and the possible urea formation by ureide degradation”.

I revised the experimental design as follows:

Line 736-738: The experiment was replicated from four plants, and all the treated-plants were cultivated simultaneously under the same conditions. Data in figures were presented as the average values with standard error, and the statistical analysis was performed using Tukey’s test.

The English language in general terms is correct, even though there are some mistakes that will be detailed below.

I asked English edition, however, I am sorry that there remain some mistakes. I revised grammatic errors as possible as I could.

The introduction gives enough background, but in some aspects it is too long, and in some others some references are missing. For example, as the main conclusion is related to ureide degradation to urea, I missed the references about two enzymes that were purified from legumes and that were characterised initially as ureidoglycolate urea-lyases (Muñoz et al, 2001 and 2006), even though they seem to catalyse another reaction that eventually could produce urea (Muñoz et al, 2010). The final paragraph is a bit weird because generally is summarizing and introducing what the authors did; however, the description that they did could be more the beginning of the Results section. I think that only the sentences between lines 160 and 163 in this paragraph should be kept in the introduction.

Thank you very much for introducing very important papers by Muñoz et al. I added these references and discussed about them. Also, I revised as your advice.

The way of describing the results is very repetitive with “Figure X shows…”. I think the authors should be a bit more imaginative about the writing in order to not to be boring in the description.

I am sorry for the repetitive expression as “Figure X shows…”. I deleted this sentence and revised it.

Perhaps I missed it, but I could not find the supplementary figures of results and the heatmap in the figure 8A (line 589) referred in the discussion. In the figure 4A, some letters for the Tukey’s test are missing.

I am very sorry missing the supplemental figures in the text. I added all the supplemental figures in the revised text.

Thank you very much for some letters that are missing in Figure 4A(b). I added the letters for this figure.

The Discussion section is very descriptive as well, and a good part of it is a repetition of the results. There are just a few references to other papers related to this. I think it could be richer if the authors had compared their results with those from some other papers (either similar or dissimilar). Furthermore, the figures referred in this section are not corresponding with the actual ones.

I added some reference papers and revised the discussion part.

The Materials and methods section is, in general, well described; however, it would be good to describe the way that the fine powder was obtained (line 664). As I noted before, I do not know if the replicates the authors talk about (line 674) are actual biological replicates (plants grown independently four times), or four samples from a unique experiment.

I revise the manuscript on your suggestions.

The Conclusions section is mainly a summary of the obtained results that could be good for the actual summary. Only the final sentence could be a conclusion.

Thank you very much for your comments. I revised the conclusion part with some future requirements.

The Abbreviations section is too long. I think it is not necessary to put the three-letter code for amino acids or some others that are in the text, as DAP.

I deleted the abbreviation list, and I use abbreviations like glutamine (Gln) for all the abbreviations.

Minor points

- I think that the authors may have something to acknowledge to someone. Please, put it in the corresponding section.

Thank you very much. I added our acknowledgment to financial support for Grant-in-Aid from JSPS.

- In the Bibliography section, there are some references that do not follow the standard pattern followed in most of them (some with pp. for the pages and some others without it). In number 6 the year is missing.

I revised the reference parts following the IJMS style. From their style, Journal pages don’t show pp., but the Book chapter needs pp. before page numbers.

- There are some references in the body text that are in a different format from numbers (line 620).

I changed the format.

- Line 445: Instead of using xX, I think it would have been better to use X-fold all over the paper.

Thank you very much for your valuable advice. I changed x X to X-fold for the indicator of magnifications.

- Please, have a look and correct some grammar mistakes that are scattered in the body text.

I am sorry for the grammar mistakes. I revised them.

Thank you very much for spending your valuable time revising our manuscript.

We appreciate your excellent comments for us.

Round 2

Reviewer 1 Report

I would like to thank the authors for revising the manuscript. The authors addressed correctly all my previous concerns; however, I think that discussion section has still to be improved. Authors should discuss better how they interpreted the results in the perspective of the working hypotheses and other published studies. Moreover, the discussion is not directed toward a broadest context. I suggest authors to remove the subsections and to link better all the paragraphs of the discussion section in a more flowing narration.

Additional comment:

Supplementary figures can be removed from the main file. 

Author Response

Dear reviewer

Thank you very much for your kind and appropriate comments and suggestions for our manuscript. We revised the manuscript with your comments.

Reviewer's Comments and Suggestions for Authors

I would like to thank the authors for revising the manuscript. The authors addressed correctly all my previous concerns; however, I think that discussion section has still to be improved. Authors should discuss better how they interpreted the results in the perspective of the working hypotheses and other published studies. Moreover, the discussion is not directed toward a broadest context. I suggest authors to remove the subsections and to link better all the paragraphs of the discussion section in a more flowing narration.

Thank you very much for your comments. I am sorry that the discussion section was not fully revised, and thank you very much for your suggestion to remove the subsections from the discussion section. I deleted subsections and revised the discussion part.

Additional comment:

Supplementary figures can be removed from the main file. 

I moved supplementary figures from main text to the end of the text.

Reviewer 2 Report

Thank you very much for your kind and constructive comments and suggestions for our manuscript.

First of all, I am very sorry that all supplementary figures are missing in the text. I submitted them as a separate file, but I should have attached them in the text.

I attached all the supplementary figures in the text.

I revised following your advice as below, and the revised parts were heightened by a yellow background.

This paper describes the effects on different metabolite levels of nodulated soybean plants of supplementing them with different forms of nitrogen (specifically nitrate, ammonium and urea). The main conclusion they seem to obtain is that ureide degradation in the different organs yields urea, and specially they remark that, for the first time, this is described in underground tissues. In general terms the paper is very descriptive (as it is the title, which I think it should be more conclusive), and from an experiment (I do not see clearly the number of used biological replicates) the authors measure all the different metabolites related with nitrogen, carbon, phosphorus, etc. However, from all the measurements carried out, only a few conclusions are withdrawn. Perhaps a deeper analysis of the same results could help to get more conclusions.

I changed the title “Changes in metabolite concentrations in the xylem sap and organs of soybean plants (Glycine max (L.) Merr.) supplied with chemical nitrogen, and the possible urea formation by ureide degradation”.

I revised the experimental design as follows:

Line 736-738: The experiment was replicated from four plants, and all the treated-plants were cultivated simultaneously under the same conditions. Data in figures were presented as the average values with standard error, and the statistical analysis was performed using Tukey’s test.

I think the title is still very descriptive and does not show any conclusion, as it is in the rest of the paper where no clear conclusions are withdrawn from the huge amount of results that they obtained.

The English language in general terms is correct, even though there are some mistakes that will be detailed below.

I asked English edition, however, I am sorry that there remain some mistakes. I revised grammatic errors as possible as I could.

The introduction gives enough background, but in some aspects it is too long, and in some others some references are missing. For example, as the main conclusion is related to ureide degradation to urea, I missed the references about two enzymes that were purified from legumes and that were characterised initially as ureidoglycolate urea-lyases (Muñoz et al, 2001 and 2006), even though they seem to catalyse another reaction that eventually could produce urea (Muñoz et al, 2010). The final paragraph is a bit weird because generally is summarizing and introducing what the authors did; however, the description that they did could be more the beginning of the Results section. I think that only the sentences between lines 160 and 163 in this paragraph should be kept in the introduction.

Thank you very much for introducing very important papers by Muñoz et al. I added these references and discussed about them. Also, I revised as your advice.

Under my point of view the introduction is still very long. Some parts could be directly removed, like the sentences between the lines 42 and 47, and some others. Other parts should have been summarized to avoid to be so long. I appreciate that the authors included the references I indicated, but they did not do it in a proper way, since these enzymes seemed to be ureidoglycolate urea-lyases at the beginning, but later it was demonstrated that they were catalysing another reaction that eventually could produce urea (Muñoz et al., 2010). Please, do it correctly. Finally, regarding the introduction I think that the initial paragraph that the authors included now should be the final paragraph that I was missing for the end of the introduction. Please, put it there.

The way of describing the results is very repetitive with “Figure X shows…”. I think the authors should be a bit more imaginative about the writing in order to not to be boring in the description.

I am sorry for the repetitive expression as “Figure X shows…”. I deleted this sentence and revised it.

I realised the authors put the former final paragraph of the introduction as the beginning of the results section, as I indicated. They also changed the way of describing the results, but now the repetitive sentence is “The average X concentration in…”. Please, there are many sentences that could be used.

Perhaps I missed it, but I could not find the supplementary figures of results and the heatmap in the figure 8A (line 589) referred in the discussion. In the figure 4A, some letters for the Tukey’s test are missing.

I am very sorry missing the supplemental figures in the text. I added all the supplemental figures in the revised text.

Thank you very much for some letters that are missing in Figure 4A(b). I added the letters for this figure.

The authors corrected these two details, but I had to download the supplementary file to see the figure legends.

The Discussion section is very descriptive as well, and a good part of it is a repetition of the results. There are just a few references to other papers related to this. I think it could be richer if the authors had compared their results with those from some other papers (either similar or dissimilar). Furthermore, the figures referred in this section are not corresponding with the actual ones.

I added some reference papers and revised the discussion part.

I think this is the main problem of the paper. The authors did not change substantially the discussion, and I think that in this section they should do an effort to interpret and relate all the different results they obtained between them and with the results in other similar papers. Instead of that, they just comment a bit the obtained results.

The Materials and methods section is, in general, well described; however, it would be good to describe the way that the fine powder was obtained (line 664). As I noted before, I do not know if the replicates the authors talk about (line 674) are actual biological replicates (plants grown independently four times), or four samples from a unique experiment.

I revise the manuscript on your suggestions.

With the suggestions made by the other reviewer this section looks pretty better now.

The Conclusions section is mainly a summary of the obtained results that could be good for the actual summary. Only the final sentence could be a conclusion.

Thank you very much for your comments. I revised the conclusion part with some future requirements.

I think this section almost remains the same as it was at the beginning, as a summary of results. The future perspectives are a good point.

The Abbreviations section is too long. I think it is not necessary to put the three-letter code for amino acids or some others that are in the text, as DAP.

I deleted the abbreviation list, and I use abbreviations like glutamine (Gln) for all the abbreviations.

Minor points

- I think that the authors may have something to acknowledge to someone. Please, put it in the corresponding section.

Thank you very much. I added our acknowledgment to financial support for Grant-in-Aid from JSPS.

OK.

- In the Bibliography section, there are some references that do not follow the standard pattern followed in most of them (some with pp. for the pages and some others without it). In number 6 the year is missing.

I revised the reference parts following the IJMS style. From their style, Journal pages don’t show pp., but the Book chapter needs pp. before page numbers.

  1. The reference to Atkins et al. in line 124 should be the number 41.

- There are some references in the body text that are in a different format from numbers (line 620).

I changed the format.

OK.

- Line 445: Instead of using xX, I think it would have been better to use X-fold all over the paper.

Thank you very much for your valuable advice. I changed x X to X-fold for the indicator of magnifications.

OK.

- Please, have a look and correct some grammar mistakes that are scattered in the body text.

I am sorry for the grammar mistakes. I revised them.

Thank you very much for spending your valuable time revising our manuscript.

We appreciate your excellent comments for us.

OK.

Author Response

Thank you very much for your kind and constructive comments and suggestions for our manuscript again.

Reviewer's Comments

I think the title is still very descriptive and does not show any conclusion, as it is in the rest of the paper where no clear conclusions are withdrawn from the huge amount of results that they obtained.

 I changed the title “Application of nitrate, ammonium, or urea changes ureides, urea, amino acids and other metabolites concentrations in xylem sap and in the organs of soybean plants (Glycine max (L.) Merr.)”.

The English language in general terms is correct, even though there are some mistakes that will be detailed below.

I am sorry for the bad English. I asked the English edition of MDPI editing, and I revised grammatic errors.

Under my point of view the introduction is still very long. Some parts could be directly removed, like the sentences between the lines 42 and 47, and some others. Other parts should have been summarized to avoid to be so long.

I deleted many part of introduction including lines 42-47 and others as your suggestion. Thank you very much. I deleted two references cited between line 42 and 47.

I appreciate that the authors included the references I indicated, but they did not do it in a proper way, since these enzymes seemed to be ureidoglycolate urea-lyases at the beginning, but later it was demonstrated that they were catalysing another reaction that eventually could produce urea (Muñoz et al., 2010). Please, do it correctly.

I added the reference by Muñoz et al., (2010).

Finally, regarding the introduction I think that the initial paragraph that the authors included now should be the final paragraph that I was missing for the end of the introduction. Please, put it there.i

I changed the final paragraph at the end of introduction.

I realised the authors put the former final paragraph of the introduction as the beginning of the results section, as I indicated. They also changed the way of describing the results, but now the repetitive sentence is “The average X concentration in…”. Please, there are many sentences that could be used.

I revised as you recommended.

The Discussion section is very descriptive as well, and a good part of it is a repetition of the results. There are just a few references to other papers related to this. I think it could be richer if the authors had compared their results with those from some other papers (either similar or dissimilar). Furthermore, the figures referred in this section are not corresponding with the actual ones.

I added some reference papers and revised the discussion part.

I think this is the main problem of the paper. The authors did not change substantially the discussion, and I think that in this section they should do an effort to interpret and relate all the different results they obtained between them and with the results in other similar papers. Instead of that, they just comment a bit the obtained results.

I am sorry for the discussion part was not fully revised in the former revision.

I fully revised the discussion part, delete the subtitle, and delete the part with anions, etc. I concentrated the discussion on the urea formation from ureides, and amino acid concentration changes.

The Materials and methods section is, in general, well described; however, it would be good to describe the way that the fine powder was obtained (line 664). As I noted before, I do not know if the replicates the authors talk about (line 674) are actual biological replicates (plants grown independently four times), or four samples from a unique experiment.

I revise the manuscript on your suggestions.

The Conclusions section is mainly a summary of the obtained results that could be good for the actual summary. Only the final sentence could be a conclusion.

Thank you very much for your comments. I revised the conclusion part with some future requirements.

Thank you very much for spending your valuable time revising our manuscript.

We appreciate your excellent comments for us.

Round 3

Reviewer 2 Report

Thank you very much for your kind and constructive comments and suggestions for our manuscript again.

Reviewer's Comments

I think the title is still very descriptive and does not show any conclusion, as it is in the rest of the paper where no clear conclusions are withdrawn from the huge amount of results that they obtained.

 I changed the title “Application of nitrate, ammonium, or urea changes ureides, urea, amino acids and other metabolites concentrations in xylem sap and in the organs of soybean plants (Glycine max (L.) Merr.)”.

I think grammatically would be better “Application of nitrate, ammonium or urea changes the concentrations of ureides, urea, amino acids and other metabolites in xylem sap and in the organs or soybean plants (Glycine max (L.) Merr.).

The English language in general terms is correct, even though there are some mistakes that will be detailed below.

I am sorry for the bad English. I asked the English edition of MDPI editing, and I revised grammatic errors.

OK.

Under my point of view the introduction is still very long. Some parts could be directly removed, like the sentences between the lines 42 and 47, and some others. Other parts should have been summarized to avoid to be so long.

I deleted many part of introduction including lines 42-47 and others as your suggestion. Thank you very much. I deleted two references cited between line 42 and 47.

OK.

I appreciate that the authors included the references I indicated, but they did not do it in a proper way, since these enzymes seemed to be ureidoglycolate urea-lyases at the beginning, but later it was demonstrated that they were catalysing another reaction that eventually could produce urea (Muñoz et al., 2010). Please, do it correctly.

I added the reference by Muñoz et al., (2010).

OK.

Finally, regarding the introduction I think that the initial paragraph that the authors included now should be the final paragraph that I was missing for the end of the introduction. Please, put it there.i

I changed the final paragraph at the end of introduction.

OK.

I realised the authors put the former final paragraph of the introduction as the beginning of the results section, as I indicated. They also changed the way of describing the results, but now the repetitive sentence is “The average X concentration in…”. Please, there are many sentences that could be used.

I revised as you recommended.

The Discussion section is very descriptive as well, and a good part of it is a repetition of the results. There are just a few references to other papers related to this. I think it could be richer if the authors had compared their results with those from some other papers (either similar or dissimilar). Furthermore, the figures referred in this section are not corresponding with the actual ones.

I added some reference papers and revised the discussion part.

Essentially the discussion part did not change.

I think this is the main problem of the paper. The authors did not change substantially the discussion, and I think that in this section they should do an effort to interpret and relate all the different results they obtained between them and with the results in other similar papers. Instead of that, they just comment a bit the obtained results.

I am sorry for the discussion part was not fully revised in the former revision.

I fully revised the discussion part, delete the subtitle, and delete the part with anions, etc. I concentrated the discussion on the urea formation from ureides, and amino acid concentration changes.

The Materials and methods section is, in general, well described; however, it would be good to describe the way that the fine powder was obtained (line 664). As I noted before, I do not know if the replicates the authors talk about (line 674) are actual biological replicates (plants grown independently four times), or four samples from a unique experiment.

I revise the manuscript on your suggestions.

OK.

The Conclusions section is mainly a summary of the obtained results that could be good for the actual summary. Only the final sentence could be a conclusion.

Thank you very much for your comments. I revised the conclusion part with some future requirements.

Thank you very much for spending your valuable time revising our manuscript.

We appreciate your excellent comments for us.